# Mutant Kras as a Biomarker Plays a Favorable Role in FL118-Induced Apoptosis, Reactive Oxygen Species (ROS) Production and Modulation of Survivin, Mcl-1 and XIAP in Human Bladder Cancer

**DOI:** 10.3390/cancers12113413

**Published:** 2020-11-18

**Authors:** Sreevidya Santha, Xiang Ling, Ieman A. M. Aljahdali, Sailee S. Rasam, Xue Wang, Jianqun Liao, Jue Wang, Christos Fountzilas, Qingyong Li, Jun Qu, Fengzhi Li

**Affiliations:** 1Department of Pharmacology and Therapeutics, Roswell Park Comprehensive Cancer Center, Elm and Carlton Streets, Buffalo, New York, NY 14263, USA; sreevidya.gp@gmail.com (S.S.); Ieman.Aljahdali@RoswellPark.org (I.A.M.A.); Jian.Liao@RoswellPark.org (J.L.); 2Canget BioTekpharma, LLC, Buffalo, New York, NY 14203, USA; 3Department of Cellular & Molecular Biology, Roswell Park Comprehensive Cancer Center, Buffalo, New York, NY 14263, USA; 4Department of Biochemistry, Jacobs School of Medicine, University at Buffalo, State University of New York, Buffalo, New York, NY 14203, USA; saileesu@buffalo.edu; 5Department of Pharmaceutical Sciences, School of Pharmacy, University at Buffalo, State University of New York, Buffalo, New York, NY 14214, USA; xwang79@buffalo.edu (X.W.); junqu@buffalo.edu (J.Q.); 6Department of Cell Stress Biology, Roswell Park Comprehensive Cancer Center, Buffalo, New York, NY 14263, USA; 7Department of Interdisciplinary Oncology, University of Arizona Cancer Center, Phoenix Arizona, AZ 85004, USA; juewang@email.arizona.edu; 8Department of Medicine, Roswell Park Comprehensive Cancer Center, Buffalo, New York, NY 14263, USA; Christos.Fountzilas@RoswellPark.org; 9College of Pharmaceutical Science, Zhejiang University of Technology, Hangzhou 310014, China; liqy@zjut.edu.cn; 10Developmental Therapeutics (DT) Program, Roswell Park Comprehensive Cancer Center, Buffalo, New York, NY 14263, USA

**Keywords:** FL118, bladder cancer, apoptosis, survivin, Mcl-1, XIAP, mutant Kras, reactive oxygen species (ROS), proteomics analysis, human bladder tumor mouse models

## Abstract

**Simple Summary:**

FL118 is a novel orally available small molecule anticancer drug. We found that bladder cancer cells with a mutant Kras is highly sensitive to FL118-induced cell growth inhibition and cell death induction through inhibiting the anti-cancer cell death and drug resistance factors (survivin, Mcl-1, XIAP). In the Kras-mutation bladder cancer cells, FL118 can stimulate the reactive oxygen species (ROS) over-production for killing bladder cancer cells and inhibiting bladder cancer cell-established tumor growth. Elimination of mutant Kras by Kras-specific shRNA technology in mutant Kras-containing bladder cancer cell-established tumor decreased FL118 effectiveness to inhibit bladder cancer tumor growth. In this regard, mutant Kras is a potential favorable biomarker for FL118. This finding is significant because mutant Kras is known to be a formidable challenge treatment resistant factor in various types of cancer. Thus, FL118 could use mutant Kras as favorable biomarker for patient selection to carry out precision medicine.

**Abstract:**

Tumor heterogeneity in key gene mutations in bladder cancer (BC) is a major hurdle for the development of effective treatments. Using molecular, cellular, proteomics and animal models, we demonstrated that FL118, an innovative small molecule, is highly effective at killing T24 and UMUC3 high-grade BC cells, which have Hras and Kras mutations, respectively. In contrast, HT1376 BC cells with wild-type Ras are insensitive to FL118. This concept was further demonstrated in additional BC and colorectal cancer cells with mutant Kras versus those with wild-type Kras. FL118 strongly induced PARP cleavage (apoptosis hallmark) and inhibited survivin, XIAP and/or Mcl-1 in both T24 and UMUC3 cells, but not in the HT1376 cells. Silencing mutant Kras reduced both FL118-induced PARP cleavage and downregulation of survivin, XIAP and Mcl-1 in UMUC3 cells, suggesting mutant Kras is required for FL118 to exhibit higher anticancer efficacy. FL118 increased reactive oxygen species (ROS) production in T24 and UMUC3 cells, but not in HT1376 cells. Silencing mutant Kras in UMUC3 cells reduced FL118-mediated ROS generation. Proteomics analysis revealed that a profound and opposing Kras-relevant signaling protein is changed in UMUC3 cells and not in HT1376 cells. Consistently, in vivo studies indicated that UMUC3 tumors are highly sensitive to FL118 treatment, while HT1376 tumors are highly resistant to this agent. Silencing mutant Kras in UMUC3 cell-derived tumors decreases UMUC3 tumor sensitivity to FL118 treatment. Together, our studies revealed that mutant Kras is a favorable biomarker for FL118 targeted treatment.

## 1. Introduction

Bladder cancers are a heterogeneous group of neoplasms in terms of clinical behavior and response to chemotherapy [1]. Most are urothelial carcinomas, which are subdivided into non-muscle-invasive bladder cancer (NMIBC) and muscle-invasive bladder cancer (MIBC) with distinct molecular features and clinical outcome [1]. NMIBCs frequently recur and, in some cases, progress to invasive and metastatic disease, while the MIBCs frequently metastasize [2]. Therefore, this is an expensive disease to treat. For NMIBC, transurethral resection is the initial treatment followed by intravesical administration of immunomodulating or chemotherapeutic agents [3], while radical cystectomy, preceded by neoadjuvant chemotherapy in most cases, is the current standard approach to manage and treat patients with MIBCs [1,4].

Current combination chemotherapies for MIBC are based on methotrexate, vinblastine, adriamycin and cisplatin (MVAC) or gemcitabine plus cisplatin, which are highly toxic with severe side effects [5]. Although various types of targeted drugs for bladder cancer are in clinical development [4,5,6,7], it remains to be seen if the clinical trials with targeted therapies hold promise for the treatment of this disease. In addition, the prognosis of patients with MIBC has not been improved in the past two decades, and bladder cancer still is one of the most expensive cancers to treat [8,9]. The median survival of patients with metastases is 15.2 months and a 5-year survival rate of 15.3% [10]. Up to date, Erdafitinib (Balversa, an FGFR inhibitor) is the first targeted drug approved by FDA for metastatic urothelial cancer. However, there are major concerns about its shorter duration of response and toxicity profile (e.g., ocular toxicity) [11]. Therefore, the development of new and effective anti-bladder cancer therapeutic agents is still a high unmet need for patients with bladder cancer, especially for those with advanced, high-grade bladder cancer.

As mentioned early, the highly heterogeneous nature of bladder cancer is a major hurdle for the development of successful therapies. Current studies indicate that activation of fibroblast growth factor receptor 3 (FGFR3) and Ras gene mutations are frequent in NMIBCs, while MIBC develops predominantly through the inactivation of the p53, retinoblastoma (RB), phosphatase and tensin homolog (PTEN) tumor suppressor pathways [12,13,14]. Oncogenic activation of both FGFR3 and Ras genes results in the stimulation of the mitogen-activated protein kinase (MAPK) and AKT pathways [15]. However, based on our studies reported here, this is unlikely the entire mechanism of this disease.

We have initially reported on a novel anticancer small molecule named FL118 [16]. However, its detailed mechanism of action continues to be investigated to determine FL118′s breadth of efficacy against a wide variety of cancers. This report continues to build on our previously published findings (refer to the Discussion section for more information) as we further advance our understanding of whether FL118 can be an effective treatment for advanced high-grade bladder cancer. Our studies reveal that mutant Kras is a favorable biomarker that can be potentially used in the stratification of cancer patients for FL118 treatment. This finding is significant because Kras mutation was a difficult target in many types of cancer, including pancreatic and colorectal cancers.

## 2. Materials and Methods

### 2.1. Reagents and Antibodies

Antibodies for poly (ADP-ribose) polymerase (PARP) (9542S), Ras^G12V^ (14412S), Ras^G12D^ (14429S), phospho-AKT (4060S), phospho-ERK1/2 (4370S), cIAP2 (3130S), XIAP (2042S), Mcl-1 (4572S) and survivin (2803S) were purchased from Cell Signaling. The antibody for Kras (101667-T32) was purchased from Sino Biological (Southeastern, PA, USA). The antibodies for survivin were also purchases from Santa Cruz Biotechnology ((sc-10811) and from Epitomics (2565-1). Antibodies for GAPDH (sc-25778/sc-47724), AKT (sc-8312) and ERK1/2 (sc-292838) were purchased from Santa Cruz. The second antibodies of goat anti-mouse IgG (G21040) and goat anti-rabbit IgG (G21234) were purchased from Invitrogen/ThermoFisher. Antibodies for topoisomerase 1 were purchased from BD Biosciences (556597) and TopoGEN (TG2012-4). All primary antibodies were diluted in the range of 500 × to 1000 ×. In this study, secondary antibodies were diluted 5000 × to 10,000 ×. Dulbecco’s modified eagle’s medium (DMEM), penicillin–streptomycin solution, trypsin EDTA and phosphate-buffered saline (PBS) were from Mediatech, Inc. (Manassas, VA, USA); Opti-MEM was from Gibco. Lipofectamine 2000 was from Invitrogen/ThermoFisher. Fetal bovine serum (FBS) was from Atlanta Biologicals. The 3-(4,5-dimethylthiazol-2-yl)-2,5-diphenyltetrazolium bromide (MTT) was from Affymetrix. The protease inhibitor cocktail was from Sigma-Aldrich; the reactive oxygen species (ROS) assay kit was from Abcam; the BCA™ protein assay kit was from Pierce and Western lightning plus ECL was from PerkinElmer (Shelton, CT, USA).

### 2.2. Cell Culture

Human bladder cancer cell lines HT1376, T24, UMUC3, RT112 and TCCSUP were originally from ATCC and were kindly provided by Drs. Yuesheng Zhang and Candace Johnson, Roswell Park Comprehensive Cancer Center. Dr. Johnson’s lab also provided human bladder cancer cell line 253 J B-V for this study. Human colorectal cancer (CRC) cell lines SNU-C2B and HCT117 were purchased from ATCC. Cells were grown under standard culture conditions at 37 °C in a humidified atmosphere containing 5% CO_2_. Cells were cultured in DMEM supplemented with 10% FBS, 100 unit/mL of penicillin and 100 μg/mL of streptomycin. Sub-confluent cells were treated with different concentrations of FL118 or with an equivalent amount of solvent DMSO according to the experiments.

### 2.3. MTT Assay

In vitro inhibitory effects of FL118 on bladder cancer cell growth was performed using MTT assay. The human bladder cancer HT1376, UMUC-3 and T24 cells were plated in 96-well plates at a density of 5000 cells/well and allowed to adhere overnight. The next day, cells were treated with a medium containing DMSO as a vehicle or with 0.5–200 nM FL118 for 24, 48 and 72 hrs. Cell viability was determined at the end of each treatment time by 3-(4,5-dimethylthiazol-2-yl)-2,5-diphenyltetrazolium bromide (MTT) assay, as previously described [17,18]. For this, the cells were incubated with 50 µL of 0.5 mg/mL MTT solution for 4 h, followed by incubation with 150 µL of DMSO for another 30 min at 37 °C. The absorbance was measured at 570 nm in a microtiter plate reader. A blank reading was subtracted from experimental readings for final calculations.

### 2.4. Immunoblotting

Immunoblotting was performed as previously described [19,20]. Briefly, the cells were lysed in a RIPA lysis buffer containing a protease inhibitor cocktail (Sigma-Aldrich, St. Louis, MO, USA) and supernatants were collected by centrifugation at 13,000 rpm for 15 min at 4 °C. Protein quantification in each sample was performed using a BCA™ protein assay kit (Pierce, Rockford, IL, USA) with bovine serum albumin (BSA) as a standard. Equal amounts of proteins were denatured and fractionated by SDS–PAGE, and the proteins were transferred onto nitrocellulose membranes. Nonspecific binding sites were blocked by incubating the membranes in 5% nonfat milk in TBS containing 0.1% Tween-20. The membranes were probed with appropriate primary antibodies overnight at 4 °C followed by appropriate horseradish peroxidase (HRP) conjugated secondary antibody for 1 h at room temperature. Protein expression was detected by enhanced Western Lightning Plus-ECL, enhanced chemiluminescence substrate (PerkinElmer, Shelton, CT, USA). Images were acquired on a Bio-Rad ChemiDoc system by using Image lab software. Expression of GAPDH was used as an internal control for total protein-loading.

### 2.5. Lentiviral shRNA-Mediated Knockdown of Kras and Puromycin Selection

Lentiviral vectors encoding scramble control shRNA and Kras-specific shRNA, respectively, were obtained from our Institute’s shRNA Core Resource, which owns various shRNA libraries in collaboration with Open Biosystems. Lentiviral shRNA constructs were packaged in pseudo-type viral particles, as described in our previous studies [16]. Subconfluent UMUC-3 cells in 6-well plates were infected with 100 µL/well of lentiviral stock in the presence of 4 µg/mL polybrene. The control scramble shRNA or the previously identified Kras-specific shRNA [21] infected cells in the plate were grown for 24 h, and the media was replaced with fresh growth media. Cells in individual wells were then split, followed by selection with puromycin (5 µg/mL) for 3–7 days. The remaining puromycin selected cell populations specific for Kras shRNA or Control shRNA were used for the experiments.

### 2.6. Overexpression of Kras^G12V^ in HT1376 Bladder Cancer Cells

HT-1376 cells were transfected with Kras^G12V^ or control vector using Lipofectamine 2000 as per the manufacturer’s instructions. Briefly, subconfluent populations of cells were transfected with 1.5 µg of plasmid DNA in one well of the 6 well plates in Opti-MEM for 24 h, followed by recovery in serum-containing medium. After 48 h of transfection, cells were treated with either DMSO or 100 nM FL118 for an additional 48 h. The lysates were prepared using RIPA lysis buffers and used for Western blot analysis.

### 2.7. Reactive Oxygen Species (ROS) Assay

Cellular reactive oxygen species (ROS) production in the presence of FL118 was measured using a ROS kit from Abcam. The bladder cancer cell lines HT1376, T24 and UMUC-3 were seeded in a black 96-well microplate with clear bottom (20,000 cells per well) and allowed to attach overnight. The next day, the cells were treated with either DMSO as a vehicle or 100 nM FL118 in phenol red-free DMEM for different time points. At the end of treatment, cells were incubated with cell-permeant reagent 2′,7′-dichlorofluorescin diacetate (DCFDA) for 45 min at 37 °C in the dark. DCFDA is a fluorogenic dye that measures hydroxyl, peroxyl and other ROS activity within the cell. After diffusion into the cell, DCFDA is deacetylated by cellular esterases to a non-fluorescent compound, which is later oxidized by ROS into a highly fluorescent compound 2′,7′–dichlorofluorescein (DCF). Plates were read at the end of 45 min incubation on a fluorescent plate reader with excitation wavelength at 485 nm and emission wavelength at 535 nm. Blank readings were subtracted from experimental readings for final calculations.

### 2.8. Proteomics Analysis of HT1376 versus UMUC3 Cells after FL118 Treatment

HT1376 and UMUC3 bladder cancer cells were treated with and without FL118 at 20 nM for 24 h and 48 h in parallel triplicate. The obtained samples were analyzed using the Qu Lab-developed robust IonStar/MS1-based proteomics technology [22]. Briefly, through a process of protein extraction, reduction, alkylation and organic solvent precipitation and digestion of the obtained protein pellets with trypsin, 4 µg peptide samples in 4 µL from each time point sample were gone through the process of Nano LC/UHF-Orbitrap LUMOS MS analyses, as described [22]. A stringent set of criteria were employed for protein identification. The IonStar processing pipeline was used for MS1 quantification of the identified proteins from individual samples. Gene ontology (GO) annotation (quantified proteins) were analyzed using online tools of DAVID Bioinformatics Resources v6.7 (http://david.abcc.ncifcrf.gov/) [23] and/or Ingenuity Pathway Analysis (IPA, Ingenuity Systems). For GO enrichment analysis, all quantified proteins were used as background. Protein function, upstream regulator, and pathway analysis were carried out using IPA for a Core analysis. Biological and functional annotations assigned by the software were manually examined and regrouped into respective categories. Hierarchical cluster analysis (uncentered Pearson correlation distance and centroid-linkage) was analyzed using Cluster 3.0 [24] and displayed by TreeView, supporting tree-based and image-based browsing of hierarchical trees (http://www.eisenlab.org), when needed. In addition to the use of IPA for pathway analyses, we have also analyzed the changes of protein network pathways in FL118-treated and untreated HT1376 cells versus UMUC3 cells using GSEA analysis tools [25,26].

### 2.9. Human Bladder Cancer Xenograft Tumor Mouse Model for FL118 Efficacy Testing

Human subjects were not applicable to this study. All in vivo experimental studies were approved by the Institutional Animal Care and Use Committee (IACUC) and performed following the IACUC-approved mouse protocol at Roswell Park Comprehensive Cancer Center. The animal work followed our previous protocols [16]. Briefly, human bladder cancer xenograft tumors were first generated through implanting 2 × 10^6^ bladder cancer cells at the flank area of severe combined immunodeficiency (SCID) mice. After the tumors grew to 800–1200 mm^3^, they were isolated and individual experimental mice were subcutaneously implanted with 30–50 mg of non-necrotic tumor masses at the flank area of individual mice. Seven to 14 days after tumor transplantation, the implanted bladder xenograft tumors were grown to 100–150 mm^3^ (defined as day 0), and mice were randomly divided into the required groups (5 mice per group) for treatment via oral administration of FL118 or vehicle. The schedule for FL118 or vehicle treatment was weekly for 4 times (weekly × 4). The FL118 in the current study used a basic oral formulation recipe, which contained FL118 (0.5 mg/mL), hydroxypropyl-β-cyclodextrin (0.5%, *w/v*), propylene glycol (1%) and hydroxypropyl methylcellulose (2% *w/v*, viscosity 40–60 cps) in saline. The formulation process was described in detail in the published patent (PCT/US2015/0022095) [27]. The vehicle solution contained hydroxypropyl-β-cyclodextrin (0.5%, *w/v*), propylene glycol (1%) and hydroxypropyl methylcellulose (2% *w/v*, viscosity 40–60 cps) in saline without FL118. Tumor length (L) and width (W) were measured using digital vernier calipers two times per week until the end of the experimental studies. The tumor volume (v) was calculated using the formula: v = 0.5 (L × W^2^). Then, the tumor size was divided by the day 0 tumor size, as percentage tumor size versus day 0. The mean tumor volume ± standard deviation (SD) at each time point was derived from 5 mice in each group. The tumor curves were made using Microsoft Excel.

### 2.10. Statistical Analysis

The experimental data were analyzed using Microsoft Excel and expressed as the mean ± SD. The statistical significance of differences was determined by Student’s *t*-test in two groups and one-way analysis of variance among multiple groups, with a *p*-value of 0.05 or less considered as significant. * *p*-value < 0.05, ** *p*-value < 0.01, *** *p*-value < 0.001. Protein band relative intensities from various Western blot results were quantified using ImageJ software and normalized against loading controls.

## 3. Results

### 3.1. Differential Inhibitory Effects of FL118 on Viability and Growth of Bladder Cancer Cells with Distinct Ras Statuses

The high-grade bladder cancer cell lines HT1376, T24 and UMUC-3 had different oncogenic Ras statuses. While HT1376 cells had wild-type Ras, T24 and UMUC-3 cells containing mutated Hras (G12V) and mutant Kras (G12C), respectively. All of the three cell lines had mutant p53. To evaluate bladder cancer cell growth inhibition by FL118 treatment, HT-1376, T24 and UMUC-3 cells were exposed to 0.5 to 200 nM FL118 for 24–72 h; cell viability was measured using MTT assay. As shown in Figure 1, T24 with Hras G12V and UMUC-3 cells with Kras G12C mutations displayed much higher sensitivity to FL118 than the Ras wild-type HT1376 cells. While maximum cell viability inhibition obtained in the HT1376 cell line was about 40% with 200 nM concentration of FL118 (72 h, Figure 1A), the same concentration of FL118 inhibited 88% of cell viability in T24 (Figure 1B) and 98.8% of cell viability in UMUC-3 cells (Figure 1C). Significantly, the FL118 highly resistant HT1376 cell line and the FL118 highly sensitive UMUC3 cell line-established xenograft tumors were also tested in animal models and support the in vitro result (see below). Additionally, we tested additional human bladder cancer cells with wild-type Ras genes (RT112, 253J B-V, TCCSUP) and found that these cell lines without Kras/Hras/Nras mutation were always relatively resistant to FL118 treatment (Appendix A, Table 1). Kras mutation in bladder cancer was low (8%) [28]. Due to various reasons, we were unable to obtain additional bladder cell lines with Kras mutation. In contrast, the mutation rate of Kras in colorectal cancer (CRC) was at a range of 40–44.7% [29,30]. Therefore, using MTT, we determined cell sensitivity to FL118 in a couple of CRC cell lines with mutant Kras. We found that CRC cell lines with mutant Kras exhibit higher sensitivity to FL118-induced cell growth inhibition (Appendix A). Since the bladder cancer cell lines HT1376, T24 and UMUC3 were commonly used in research, we therefore mainly used these representative three cell lines for the molecular level studies elucidated in this paper.

### 3.2. Human Bladder Cancer Cells with Different Ras Statuses Exhibit Differential Sensitivity to FL118-Induced Apoptosis

We anticipated that the differential cell growth and viability inhibition by FL118 in bladder cancer cells would be associated with a differential induction of cell apoptosis. In order to determine the effect of FL118 treatment on the apoptotic potential of bladder cancer cells, we examined PARP cleavage as a hallmark of apoptosis by Western blot analyses. We found that 10–100 nM of FL118 induced strong PARP cleavage in T24 (Figure 2B, Appendix A) and UMUC-3 (Figure 2C, Appendix A) bladder cancer cells. In contrast, there was no PARP cleavage difference in HT1376 bladder cancer cells (Figure 2A, Appendix A). This result is fully consistent with the MTT assay resulted indicating that bladder cancer cells with different Ras statuses exhibited differential sensitivity to FL118 cell growth and viability inhibition (Figure 1, Table 1, Appendix A).

### 3.3. Differential Inhibition of the IAP and Bcl-2 Family Proteins by FL118 Is Associated with FL118 Sensitivity to Bladder Cancer Cells with Different Genetic Backgrounds

Our previous studies in colorectal and head-&-neck cancer cells indicated that FL118 selectively inhibits the expression of survivin, Mcl-1, XIAP and/or cIAP2, which were demonstrated to play important roles in FL118-induced cell growth inhibition and apoptosis induction tested through genetic silencing or overexpression of these genes [16,31]. We then reasoned that the differential effect of FL118 on bladder cancer cell growth inhibition and apoptosis induction might be associated with the differential inhibition of one or more of these anti-apoptotic proteins. Therefore, we determined the expression changes in the members of the IAP and the Bcl-2 family after FL118 treatment in bladder cancer cells through Western blot analyses. We found that FL118 treatment induced different levels of downregulation of the IAP and Bcl-2 family proteins. Specifically, while cIAP2 did not show significant inhibition by FL118 in any of these three cell lines (HT1376, T24, UMUC3), FL118 treatment strongly inhibited the expression of survivin, XIAP and Mcl-1 in UMUC3 cells (Figure 3C, Appendix A), and inhibited survivin and XIAP in T24 bladder cancer cells (Figure 3B, Appendix A). However, FL118 failed to inhibit these proteins in HT1376 cells (Figure 3A, Appendix A). This result is consistent with the ability of FL118 in bladder cancer cell growth inhibition and apoptosis induction shown in Figure 1 and Figure 2. Together, HT1376 cells were resistant to FL118 treatment, while UMUC-3 and T24 cells were sensitive to FL118 treatment, with UMUC-3 being the most sensitive to FL118.

### 3.4. AKT and ERK1/2 Signaling Pathways May Not Play a Major Role in FL118 Differential Sensitivity in Bladder Cancer Cells

Previous studies have indicated that the activated AKT and MAPK signaling can upregulate the expression of survivin through the AKT [32,33,34,35,36] and/or MAPK signaling transduction pathways [34,36,37,38]. It is also well known that activated Ras–MAPK and PI3K–Akt pathways promotes cancer cell survival and inhibits apoptosis through upregulation of anti-apoptotic proteins [8]. It is possible that the inhibition of survivin, XIAP and/or Mcl-1 by FL118 were due to the inhibition of AKT and/or MAPK activation by FL118. In other words, it is possible that there are high constitutively active AKT (high phosphorylation of AKT ser473) and/or ERK1/2 (high phosphorylation of Thr202/Tyr204 for human ERK1 and Thr185/Tyr187 for human ERK2) to maintain the high expression of survivin, XIAP and Mcl-1 for bladder cancer cell survival and proliferation. Therefore, we determined the effect of FL118 on the inhibition of the active/phosphorylated AKT and ERK1/2 in these bladder cancer cells. Our studies indicated that there is a high constitutively active AKT (high p-AKT) and ERK1/2 (high p-ERK1/2) in all the three bladder cancer cell lines (Figure 4, Appendix A). This is somehow inconsistent with the fact that these three cell lines exhibited differential sensitivity to FL118 treatment (Figure 1, Figure 2 and Figure 3, Table 1, Appendix A). This suggests that AKT and ERK1/2 pathways may not play a determinant role in FL118 sensitivity. Consistent with this notion, our studies indicated that FL118 treatment induced inconsistent modulation of total ERK and p-ERK and AKT and p-AKT in the three cell lines (Figure 4, Appendix A). For example, FL118 inhibited p-Akt in both HT1376 and UMUC3 cells at 24 h and 48 h time points (Figure 4A,C). However, HT1376 is resistant to FL118, while UMUC3 is highly sensitive to FL118. In contrast, FL118 significantly induced p-AKT (survival signaling) and p-ERK1/2 (growth signaling) in T24 cells (Figure 4B, Appendix A). However, T24 cells still exhibited a high sensitivity to FL118 treatment (Figure 1 and Figure 2, Table 1, Appendix A). Nevertheless, while the current studies could not rule out the possibility that FL118-mediated inhibition of p-AKT and p-ERK1/2, as well as total AKT and total ERK1/2 by FL118, may play a role in FL118 anticancer efficacy, these signaling pathways may not play the critical determinant role in FL118-mediated inhibition of bladder cancer cell viability and the induction of apoptosis. Thus, additional mechanisms appear to be involved in FL118-mediated bladder cancer cell viability inhibition and apoptosis induction.

### 3.5. Inhibition of Top1 Expression Is Independent of FL118 Sensitivity or Resistance in Bladder Cancer Cells

It has been documented that downregulation of the expression of topoisomerase I (Top1) protein by CPTs is a CPT resistance factor [39,40,41,42]. One reason for this is that Top1 inhibition by CPTs is usually associated with a topoisomerase II (Top2) activity increase [41,43]. Thus, Top2 may partially perform the Top1 work in cancer cells, which could not be controlled by CPTs. In fact, the intensity of CPT-induced downregulation of Top1 expression is positively associated with the intensity of cell resistance to CPTs. For example, CPT effectively inhibits Top1 expression in the CPT-resistant breast cancer cell line BT474, while CPT is unable to inhibit Top1 expression in the CPT-sensitive breast cancer cell line ZR75–1 [44]. Consistent with these observations, it has been reported that reduced Top1 expression and/or Top1 catalytic activity in cancer cells is associated with increased resistance to CPTs [45,46], while increased Top1 expression in cancer cells sensitizes CPTs [47,48,49]. In contrast to these findings for CPTs, the FL118-sensitive bladder cancer cell lines (T24, UMUC-3) resulted in the least Top1 expression after FL118 treatment (Figure 5B,C, Appendix A), while the FL118-resistant HT1376 cell line still had a substantial level of Top1 expression after FL118 treatment (Figure 5A). This observation is consistent with our recent finding that FL118 antitumor efficacy in colorectal cancer cells is irrelevant of Top1 expression status (regardless of Top1 either negative, low or high) [50]. Human xenograft tumors with high Top1 expression may not be sensitive to FL118 treatment, while tumors with low/negative Top1 expression can be highly sensitive to FL118 treatment [50]. Thus, our current data further demonstrated that the differential effect of FL118 on the three bladder cancer cell lines is irrelevant to Top1 expression. In other words, this Top1 molecular change cannot explain the differential sensitivity of HT1376, T24 and UMUC3 to FL118 treatment.

### 3.6. Mutant Kras in Bladder Cancer Cells Plays an Important Role in the Increase of FL118 Anticancer Cell Activity

As mentioned before, HT1376 has wild-type Ras, while T24 has mutant Hras and UMUC-3 has mutant Kras. Thus, based on the fact that HT1376 is resistant, but T24 and UMUC-3 are sensitive to FL118 treatment, it is possible that the mutation-activated Ras (mutant Ras) is one factor that contributes to FL118 high sensitivity. To determine whether mutant Ras plays a role in FL118 treatment sensitivity, we employed FL118′s most sensitive bladder cancer cell line UMUC-3 and FL118′s most resistant bladder cancer cell line HT1376 as our study model system. We first checked the PARP cleavage and the expression of XIAP, Mcl-1 and survivin as the indicators of the apoptotic potential and anti-apoptotic status after FL118 treatment of UMUC-3 cells with no transfection, with control scramble shRNA (CshRNA) transfection and with our previously identified Kras-specific shRNA (KrasshRNA [21]) transfection to silence the activated mutant Kras, respectively. We found that UMUC-3 cells became more resistant to FL118 treatment-induced apoptosis (PARP cleavage) after the silencing of mutant Kras (Figure 6A, Appendix A). Furthermore, silencing of mutant Kras made FL118 lose or partially lose the ability to inhibit the expression of XIAP, Mcl-1 and survivin (Figure 6A, Appendix A). We were able to further determine the effect of mutant Kras silencing on FL118-inhibited UMUC-3 cell viability. Consistently, the silencing of mutant Kras decreased FL118′s ability to inhibit UMUC-3 cell viability (Figure 6B). This suggests that mutant Kras play an important role in increasing FL118 anticancer efficacy.

Next, we performed the alternative experiment to determine the effect of overexpression of mutant Kras G12V in FL118-induced apoptosis level in the FL118-resistant HT1376 cells. We found that after forced expression of the constitutive active Kras G12V in HT1376 cells, FL118 treatment showed an increase in PARP cleavage levels (Figure 6C, Appendix A). However, we repeated this experiment a couple of times, and the result is not very strong. This could be due to multiple reasons, including transfection efficiency and/or endogenous wild-type Kras interferences, etc. Additionally, for this overexpression of the Kras G12V experiment, we were unable to find FL118-mediated inhibition of anti-apoptotic proteins, such as XIAP and Mcl-1 (Figure 6C, Appendix A).

### 3.7. FL118 Treatment Increases the Generation of Reactive Oxygen Species (ROS) in Ras-Mutated Bladder Cancer Cells

Previous studies reported that mutant Kras^G12V^ (K-ras V12) promotes the reactive oxygen species (ROS) production in colorectal cancer cells through activation of the p38-PDPK1-PKCδ/p47^phox^-NOX1 cascade [51], which promotes malignant transformation [51,52]. Therefore, it is possible that FL118 takes advantage of this pathway to promote ROS overproduction to kill cancer cells. Based on this rationale, we measured whether FL118 treatment increases ROS production in bladder cancer cells. In order to measure the effect of FL118 on ROS generation in bladder cancer cells, we used a cell-permeable reagent 2′,7′-dichlorofluorescin diacetate (DCFDA), a fluorogenic dye that measures hydroxyl, peroxyl and other ROS activity within the cell. After diffusion into the cell, DCFDA is deacetylated by cellular esterases to a non-fluorescent compound, which is later oxidized by ROS into 2′,7′-dichlorofluorescein (DCF), giving out fluorescence. Our studies revealed that FL118 did not increase ROS production in the FL118-resistant HT1376 cells (Figure 7A, left panel). In contrast, FL118 significantly increased ROS production in the two FL118-sensitive T24 and UMUC-3 bladder cancer cells (Figure 7A, middle and right panels). Furthermore, the silencing of mutant Kras significantly abrogated FL118-mediated ROS production (Figure 7B). Specifically, we used the UMUC-3 cells as a model system to knock down mutant Kras using Kras-specific shRNA. We found that silencing of mutant Kras significantly reduces ROS production induced by FL118 treatment (Figure 7B). Together, based on the result shown in Figure 7A,B, we conclude that FL118 takes advantage of mutant Kras-mediated ROS generation to further increase ROS overproduction as one-way to kill cancer cells.

### 3.8. Proteomics Analysis Indicates That FL118 Treatment Induced a Profound and Opposed Kras-Relevant Protein Signaling Changes in UMUC3 Cells versus in HT1376 Cells

We hypothesized that if the mutant Kras plays an important role in the FL118 antitumor activity, we should find a differential Kras signaling-relevant downstream protein change in UMUC3 cells versus in HT1376 cells after FL118 treatment. In this regard, we performed a state-of-the-art proteomics analysis of global protein changes after FL118 treatment for 24 h and 48 h in UMUC3 cells versus in HT1376 cells. The proteomics analysis has identified a total of 5902 proteins (Appendix A). From the 5902 proteins, we further sorted and identified the proteins that are relevant to the Kras signaling pathway (Appendix A). Through analyses of all the 615 Kras signaling pathway-associated proteins (Appendix A), we identified 70 proteins that exhibited a significant decrease (*p* value < 0.05) in FL118-treated UMUC3 cells, while these proteins in HT1376 cells either increased or had no significant change after FL118 treatment (Appendix A). In parallel, we also identified 67 proteins that exhibited significant increase (*p* value < 0.05) in FL118-treated UMUC3 cells, while these proteins in HT1376 cells either decrease or have no significant change after FL118 treatment (Appendix A). Based on the function of the total 137 (70 + 67) Kras signaling pathway-relevant proteins (Appendix A), we further classified them into different classes (Appendix A). In order to quickly observe the expression behavior difference of these proteins in each class, the data sets were presented in heatmap and histogram in Figure 8 and Appendix A, respectively. These studies revealed that FL118 treatment induced a profound and opposing Kras signaling pathway-relevant signaling protein change in UMUC3 cells versus in HT1376 cells.

### 3.9. UMUC3, but Not HT1376 Bladder Cancer Cell-Derived Xenograft Tumor Exhibits High Sensitivity to FL118 Treatment in Animal Models

Our in vitro experimental data demonstrated that HT1376 bladder cancer cells are highly resistant to FL118 treatment, while UMUC-3 bladder cancer cells are highly sensitive to FL118 treatment in terms of (1) cell growth/viability inhibition (Figure 1, Appendix A), (2) apoptosis induction (Figure 2, Appendix A), (3) anti-apoptotic protein inhibition (Figure 3, Appendix A), (4) the role of Kras status (Figure 6, Appendix A) and (5) ROS production (Figure 7). Consistently, our proteomics data also indicated a profound and opposing modulation of mutant Kras signaling pathway-relevant proteins (Figure 8, Appendix A). Therefore, it is very important to determine whether the in vitro experimental outcome from FL118 treatment could be mimicked and be consistent with the in vivo outcomes from the HT1376 and UMUC3 bladder cancer cell-derived xenograft tumor in SCID mice. In this regard, we first established HT1376 and UMUC3 bladder cancer xenograft tumors in SCID mice; then, we used these tumor models and determined the FL118 efficacy to inhibit the bladder cancer xenograft tumor growth. As shown, consistent with the in vitro experimental results, UMUC-3 tumors grown in SCID mice exhibited high sensitivity to FL118 treatment (Figure 9A); that is, FL118 can eliminate UMUC3 tumors after three-time treatment (Figure 9A). In contrast, HT1376 tumors grown in SCID mice exhibited high resistance to FL118 treatment (Figure 9B); that is, FL118 was basically unable to inhibit HT1376 tumor growth (Figure 9B). Furthermore, FL118 was less effective in inhibiting the UMUC3 tumors with the partial silencing of mutant Kras (Figure 9E). Together, the in vivo data related to the FL118 efficacy is fully consistent with the in vitro molecular and cell cultural studies with FL118.

## 4. Discussion

In order to facilitate a better discussion of the studies reported in this paper, we must first briefly introduce the discovery and the previous findings on FL118. We have generated the survivin promoter-driven luciferase reporter-stably expressed cancer cell models from the colon (HCT116), lung (A549), breast (MCF-7), prostate (PC-3) and ovary (2008) cancer cells [53]. Using these genetically engineered cancer cell models through high throughput screening (HTS) of compound libraries, followed by hit-to-lead in vitro and in vivo testing, we identified the novel small molecule anticancer drug FL118 [16]. We found that FL118 inhibits multiple anti-apoptotic proteins in the inhibitor of apoptosis (IAP) and Bcl-2 families (survivin, Mcl-1, XIAP, cIAP2) [16] as well as other oncogenic proteins such as MdmX/Mdm4 in the p53 signaling pathway [54]. Furthermore, while irinotecan/SN-38 (the active metabolite of irinotecan) and topotecan are substrates of the drug efflux pump proteins ABCG2/BCRP [55,56,57,58,59,60] and P-gp/MDR1 [61,62,63,64,65,66], FL118 is not a substrate of ABCG2 and P-gp, and can overcome treatment resistance resulting from the expression of ABCG2 [67] or P-gp [68]. This might be one of the reasons that FL118 can effectively overcome irinotecan and topotecan resistance [68] and can be orally available. Recent studies also revealed that FL118 targets drug-resistant pancreatic cancer cells, cancer stem cells and exhibits exceptional efficacy for the inhibition of pancreatic cancer patient-derived xenograft (PDX) tumors in animal models [69]. Additionally, in terms of the feature of FL118 acting as a non-efflux pump protein substrate, the use of non-efflux pump (e.g., ABCG2) substrate drugs, instead of combination with an efflux pump protein inhibitor, is a new trend in the field for anticancer drug development [70]. In this regard, FL118 is a small molecule drug that is well matched to this trend.

The current targeted agents for human bladder cancer treatment have shown limited efficacy with substantial side effects. Mutations of key genes downstream of the drug target is a key resistance mechanism and a major obstacle to the development of efficacious targeted therapies [8]. In this study, we report the differential sensitivity of high-grade human bladder cancer cell lines with mutant Hras (T24) or with mutant Kras (UMUC-3) versus wild-type Ras (HT1376) to FL118. Intriguingly, our studies revealed that the UMUC-3 and T24 bladder cancer cells with Kras and Hras mutations were highly sensitive to FL118 treatment; in contrast, HT1376 cells without Ras mutation showed resistance to FL118 in the cancer cell viability assay after FL118 treatment (Figure 1). Additional bladder cancer cell lines with wild-type Ras also exhibited much less sensitivity to FL118 treatment in comparison with bladder cancer or CRC cells with mutant Ras (Table 1, Appendix A). Such intriguing observation provides a possibility that mutation of Ras (Hras or Kras in this study) may represent a favorable predictive biomarker for FL118-targeted precision medicine. In order to validate this innovative concept, we performed a series of additional experiments to gain insight into the molecular events and mechanism of action for FL118 using the FL118-sensitive T24 and UMUC-3 bladder cancer cells versus the FL118-insensitive HT1376 bladder cancer cells. Consistent with the effect of FL118 on bladder cancer cell viability (Figure 1), FL118 was able to strongly induce PARP cleavage in both T24 and UMUC-3 cells but failed to induce PARP cleavage in HT1376 cells (Figure 2). Furthermore, the differential effect of FL118 on apoptosis induction in UMUC-3 and T24 cells versus in HT1376 cells were supported by our additional studies that FL118 was able to inhibit the expression of multiple anti-apoptotic proteins in T24 cells (survivin, XIAP) and UMUC-3 (survivin, XIAP, Mcl-1), while FL118 failed to inhibit any of these proteins in HT1376 cells (Figure 3). This is consistent with the previous findings that the inhibition of survivin, Mcl-1, XIAP and cIAP2 by FL118 plays a role in FL118 antitumor efficacy [16,31].

Previous studies demonstrated that activation of the AKT [71,72,73] and ERK [73,74,75] signaling is involved in mediating drug resistance and also linked to the upregulation of anti-apoptotic proteins such as XIAP [76]. We previously also found that paclitaxel-induced survivin expression is through activation of the PI3K-AKT and MEK-ERK1/2 pathways as the paclitaxel-resistant factor, and the inhibition of these two pathways by their specific inhibitors decreased survivin expression and sensitized paclitaxel efficacy in breast cancer cells [20]. Therefore, we investigated the effect of FL118 on the modulation of the total and phosphorylated AKT and ERK1/2 over time. We found that all of the three bladder cancer cell lines (HT1376, T24, UMUC-3) have the high constitutive expression and phosphorylation (activation) of AKT (p-Akt on ser473) and ERK1/2 (p-ERK1 on Thr202/Tyr204 and p-ERK2 on Thr185/Tyr187) (Figure 4, Appendix A). However, modulation of the total AKT and p-Akt, as well as the total ERK1/2 and p-ERK1/2 by FL118 treatment (Figure 4, Appendix A), is not fully consistent with the data obtained in the rest of our studies shown in Figure 1 (FL118 effects on cell viability), Figure 2 (FL118 effects on cell apoptosis) and Figure 3 (FL118 effects on anti-apoptotic protein expression), while the data from Figure 1, Figure 2 and Figure 3 are highly consistent with each other as discussed early. Therefore, we hypothesized that alternative mechanisms of action and signaling pathways for FL118′s antitumor activity should exist in bladder cancer cells.

FL118 is structurally similar to topotecan and irinotecan, which are FDA-approved camptothecin (CPT) analogs for cancer treatment. However, the high efficacy of FL118 antitumor ability is irrelevant to CPTs’ therapeutic target, Top1 as FL118 can exhibit high antitumor efficacy to tumors that have low or negative Top1 expression [50]. As mentioned earlier, this is in contrast to the reported finding that reduced Top1 expression and/or Top1 catalytic activity in cancer cells is associated with increased resistance to CPTs [45,46] and that the increased Top1 expression in cancer cells sensitizes CPTs [47,48,49]. Studies have demonstrated that downregulation of Top1 protein by CPT is a CPT resistance mechanism [39,40,41,42] and that the intensity of CPT-induced downregulation of Top1 expression is positively associated with the intensity of cell resistance to CPTs [44]. In opposition to these previously documented findings for CPTs, our studies with bladder cancer cells as presented in this report revealed that FL118 has the least ability to inhibit Top1 expression in the resistant cell line HT1376 (Figure 5A), while FL118 has a better ability to inhibit Top1 expression in the FL118-sensitive T24 and UMUC-3 cells (Figure 5B,C). Thus, downregulation of Top1 expression is not a resistant factor to FL118, and Top1 could not be used as a biomarker to predict treatment resistance and sensitivity for FL118. Actually, it should be noted that growing evidence has revealed that CPT and CPT analogs can also have Top1-independent activity to inhibit cancer cells [77] or neutralize HIV-1 [78]. More examples of Top1-independent cases were recently reviewed [79].

Identification of a biomarker for predicting the response to FL118 treatment is important for personalized treatment decisions. We next used UMUC-3 cells as a model system to test whether mutant Kras plays a positive role in FL118 antitumor activity. Our studies revealed that silencing mutant Kras in UMUC-3 cells decreases FL118′s ability to induce apoptosis (PARP cleavage) and at least partially abrogates FL118-mediated inhibition of the expression of XIAP, Mcl-1 and survivin (Figure 6A). However, genetic overexpression of Kras G12V in HT1367 cells resulted in no expression changes of XIAP and Mcl-1 while slightly increasing FL118-induced PARP cleavage (Figure 6C). One simple explanation for the minimal effect of forced expression of Kras G12V on FL118 function could be due to low transfection efficiency and/or alternatively, additional protein factors in the HT1376 cells may be involved in FL118-mediated inhibition of multiple anti-apoptotic proteins. This remains to be investigated in the future.

Consistent with the result derived from in vitro studies, UMUC3-established xenograft tumors exhibited high sensitivity to FL118 (Figure 9A), while HT1376-established xenograft tumors exhibited high resistance to FL118 (Figure 9B). Furthermore, Kras shRNA silencing of mutant Kras in the UMUC3 cell-generated xenograft tumor showed less sensitivity to FL118 treatment in comparison with the UMUC3 tumors manipulated with control shRNA (Figure 9E). We noticed that the effective difference is relatively small. This likely contributes to the stably mutant Kras-silencing UMUC3 cells only exhibiting a moderate decrease of mutant Kras expression (i.e., a moderate expression of mutant Kras was still there) in the studies (Figure 9D). Nevertheless, these observations indicate that mutant Kras plays a favorable role in FL118 anti-bladder cancer cell activity and could be used as a biomarker for bladder cancer patient selection for FL118 precision medicine. This finding is highly innovative and significant since Ras mutation, such as Kras mutation, is a well-known major treatment-resistant factor in many types of cancer, especially in pancreatic and colorectal cancers. However, we noticed that in the case of Kras^G12D^ in the pancreatic cancer HPAF-II cell line, the silencing of Kras^G12D^ using Kras shRNA could increase FL118-induced apoptosis [21], suggesting that Kras^G12D^ in HPAF-II cells is somewhat of a resistant factor for FL118. This inconsistency could result from genetic differences in other genes in the background and may not be cancer type-specific or Kras mutation-specific.

Previous studies indicated that the histone deacetylase inhibitor FR901228 selectively induces ROS production in Hras-expressing J82 bladder cancer cells versus the parental cells [80]; ROS induction can render both mutant and wild-type Kras pancreatic cancer cells susceptible to Ad.mda-7/IL-24-induced apoptosis [81]. It was also reported that Kras G12V (K-ras V12) promotes ROS production in colorectal cancer cells through activation of the p38-PDPK1-PKCδ/p47^phox^-NOX1 cascade [51], which promotes malignant transformation [52]. Additionally, ROS-mediated targeted therapies represent a new area of study for the treatment of bladder cancer [82]. In order to have a better understanding of the functional relationship of mutant Kras and mutant Hras with favorable FL118 efficacy, we first determined whether there is a differential ROS induction by FL118 treatment in the three bladder cancer cells (HT1376, T24, UMUC-3). We theorized that FL118 might take advantage of mutant Kras and mutant Hras to stimulate ROS production. As expected, FL118 treatment rapidly increased ROS production in both T24 and UMUC-3 cells, but not in HT1376 cells (Figure 7A), which is consistent with our previous observation that the plant-isolated compound AMR-MeOAc in combination with a low concentration of FL118 increases ROS production in the pancreatic cancer HPAF-II cell line [21]. We then explored the potential signaling pathway of FL118-mediated induction of ROS production through mutant Kras using the bladder cancer UMUC-3 cells as a model system. Our studies revealed that the silencing of mutant Kras by Kras-specific shRNA significantly decreased FL118-induced ROS production (Figure 7B). Thus, our studies at least partially explain why mutant Kras (G12V, G12C) can be used as a favorable biomarker for FL118 efficacy prediction.

Given the other in vitro data obtained, we next investigated the possibility that after FL118 treatment, Kras signaling pathway-relevant proteins may have expression level changes in the FL118-sensitive UMUC3 cells (mutant Kras) distinct from those in the FL118-insensitive HT1376 cells with wild-type Kras. We performed proteomic analyses of UMUC3 cells versus HT1376 cells after FL118 treatment (Appendix A). Our data indicated that many proteins associated with mutant Kras signaling pathways exhibit opposing modulations in UMUC3 versus HT1376 cells after FL118 treatment (Figure 8, Appendix A). These findings are fully consistent with our conclusion that mutant Kras is an important potential biomarker to predict cancer cell sensitivity to FL118 treatment. However, although UMUC3 cells were in an apoptotic state after FL118 treatment, while HT1376 were not, we noted a certain inconsistency between the data from Western blots and the data from proteomics. For example, for the data shown in Appendix A, at the time point of FL118 treatment for 24 h, survivin (BIRC5) decreased in UMUC3 cells and increased in HT1376 cells, which is consistent with the results obtained in the Western blots (Figure 3A,C). In contrast, for the data shown in Appendix A, at the time point of FL118 treatment for 48 h, survivin (BIRC5) increased in both UMUC3 and HT1376 cells, which is unexpected for the UMUC3 cells. We thought that since at the time point of 48 h, only a portion of the UMUC3 cells were attached, the attached UMUC3 cells used in proteomics analyses at the 48 h likely triggered a survival feedback mechanism and induced the expression of certain survival factors like survivin. Consistent with this notion, we determined the expression of survivin after 48 h FL118 treatment using the detached cells and found that the detached UMUC3 cells showed survivin decrease (Appendix A top panel). Nevertheless, as shown in Appendix A, Bax (a Bcl-2 family apoptotic protein) increased the UMUC3 cells in both FL118 treatments for 24 and 48 h time points, indicating that UMUC3 cells were in an apoptotic status, while HT1376 cells did not show increased Bax expression in both time points. It is important to note that the dynamic regulation of certain cancer cell survival factors like survivin in a dynamic manner has indicated cancer cell heterogeneity against various treatments, and this has also indicated a complicated network for cancer cell survival factors through various cell survival feedback regulations, especially in the case of survivin as a therapeutic target [83]. This situation may have well explained one interesting observation in our proteomics data shown in Appendix A. The expression modulation of certain proteins by FL118 is in a time course-dependent manner. That is, certain proteins could be upregulated by FL118 treatment at the 24 h time point and then went into another manner to be downregulated by FL118 treatment at the 48 h time point and vice versa (Figure 8 and Appendix A). Such dynamic regulation by a drug is very interesting and may include novel mechanisms that could be further explored for cancer therapeutics.

## 5. Conclusions

In this report, we have identified the FL118 differential modulation of apoptosis and the expression of anti-apoptotic proteins (survivin, XIAP, Mcl-1) in FL118-sensitive versus FL118-insensitive bladder cancer cells. We have further identified that the mutation-activated Kras (mutant Kras) is positively associated with FL118 efficacy and mechanistically acts as a mediator through which FL118 induces ROS production. Consistently, proteomics studies indicated that FL118 treatment induces a profound and opposing modulation of mutation-activated Kras signaling pathway-relevant proteins in UMUC3 cells versus in HT1376 cells. Significantly, our in vivo studies are fully consistent with the obtained molecular and cell biology data. Thus, our studies suggest that mutant Kras is a favorable biomarker for the prediction of FL118 sensitivity in bladder cancer. This finding will not only benefit a subset of bladder cancer patients with mutant Kras in their tumor but also benefit patients with other types of Kras-mutated cancers. For example, CRC and pancreatic cancers have high Kras mutation rates, and our studies indicated that FL118 has high efficacy to treat CRC cell xenograft tumors [50] and pancreatic cancer patient-derived xenograft (PDX) tumors [69]. Therefore, FL118 will first use CRC and pancreatic cancers in upcoming clinical trials. The significance of the finding in this report is that the use of mutant Kras as a biomarker for patient selection would increase patient response rates to FL118 treatment while avoiding unnecessary treatment. This would decrease costs while increasing the quality of life of these patients.

## Figures and Tables

**Figure 1 cancers-12-03413-f001:**
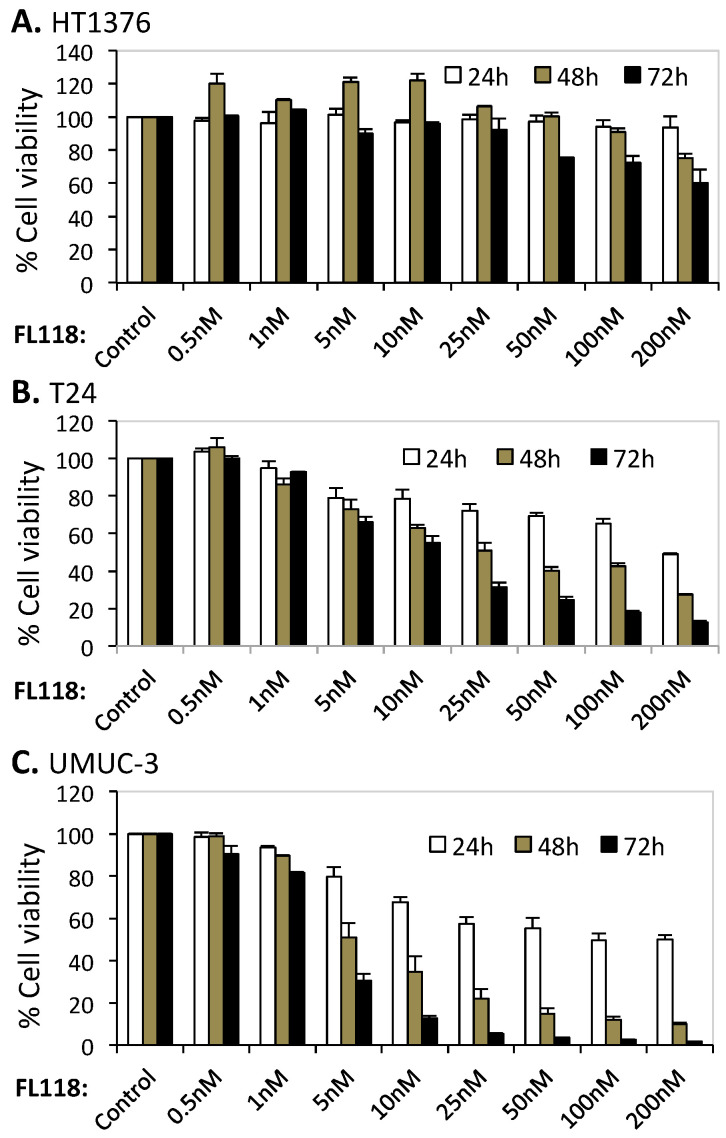
Differential inhibition of human bladder cancer cell growth by FL118. Subconfluent bladder cancer cells HT1376 (wild-type Ras), T24 (Hras G12V mutant) and UMUC-3 (Kras G12C mutant) in 96-well plates were treated with either DMSO (control) or a series of FL118 concentrations for 24 h–72 h followed by MTT assay to determine cell viability. (**A**) HT-1376 cells with wild-type Ras were insensitive to FL118 treatment compared to other cell lines (T24, UMUC-3). (**B**,**C**) Ras mutant T24 and UMUC-3 cells are highly sensitive to FL118 treatment. Each bar in the histogram shown in (**A**–**C**) is the mean ± SD from three independent assays.

**Figure 2 cancers-12-03413-f002:**
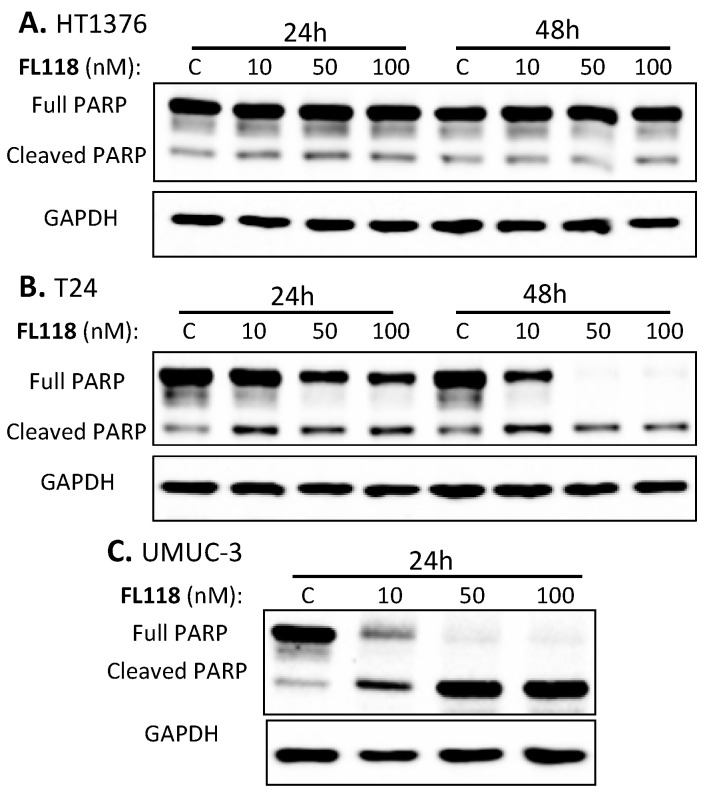
Induction of differential poly (ADP-ribose) polymerase (PARP) cleavage in bladder cancer cells by FL118. Subconfluent bladder cancer cells in 60 mm plates were treated with DMSO (as a vehicle) or with 10, 50 and 100 nM FL118 for 24 h and/or 48 h, as shown, followed by Western blot analyses to determine PARP cleavage. (**A**) HT-1376 cells with wild-type Ras show no significant PARP cleavage after 24 or 48 h FL118 treatment. (**B**) T24 cells with Hras G12V mutation showed the cleavage of PARP with all concentrations of FL118 treatment. (**C**) mutant Kras (UMUC-3) cells with Kras G12C mutation showed the maximum PARP cleavage with the disappearance of total PARP bands after 24 h FL118 treatment. GAPDH (glyceraldehyde 3-phosphate dehydrogenase) shown in A, B and C were used as internal controls of total protein-loading.

**Figure 3 cancers-12-03413-f003:**
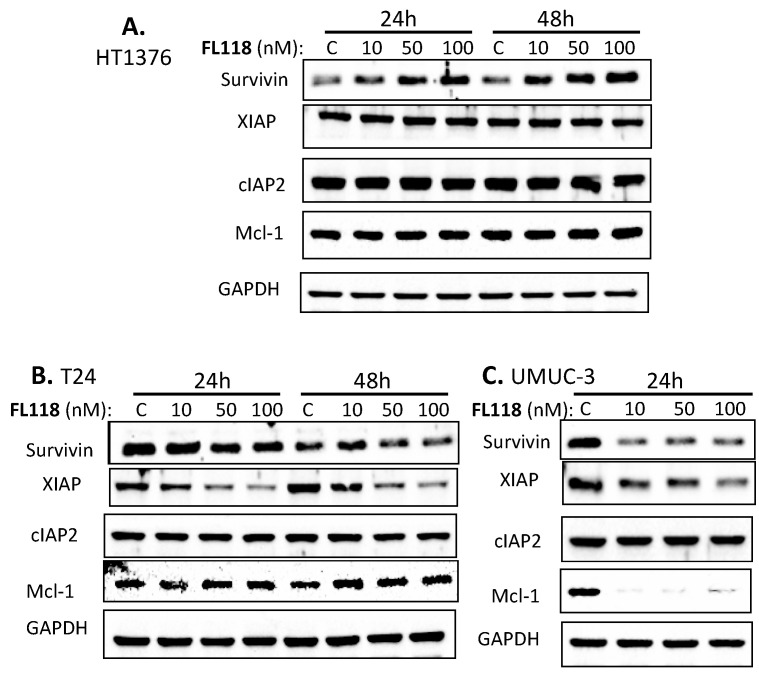
Inhibition of survivin, Mcl-1 and/or XIAP by FL118 in bladder cancer cell lines by FL118. Cells were treated with DMSO or 10–100 nM FL118 for 24 h and/or 48 h, as shown, followed by Western blot analyses to determine the expression of survivin, Mcl-1, XIAP and cIAP2. (**A**) In HT1376 cells, FL118 treatment did not inhibit any of these anti-apoptotic proteins. (**B**) T24 cells showed inhibition of XIAP and survivin after FL118 treatment. (**C**) FL118 treatment showed strong inhibition of Mcl-1, survivin and XIAP in UMUC-3 cells after FL118 treatment. GAPDH shown in A, B and C is the internal control of total protein-loading.

**Figure 4 cancers-12-03413-f004:**
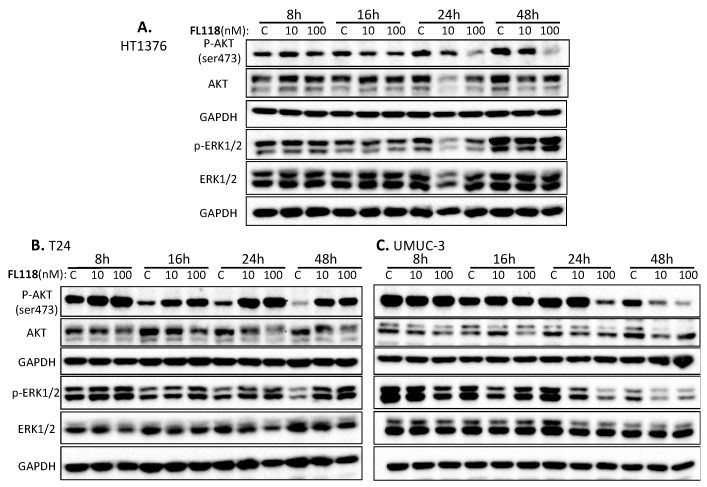
Effect of FL118 on AKT and ERK1/2 expression and phosphorylation/activation in bladder cancer cells. Subconfluent bladder cancer cells were treated with (10, 100 nM) and without (C, control/vehicle) FL118 for 8 h, 16 h, 24 h and 48 h, as shown, followed by Western blot analyses to determine the expression of AKT and ERK1/2 as well as the phospho-AKT (ser473) and phospho-p44/42 MAPK (ERK1/2) (Thr202/Tyr204). (**A**) Treatment of HT1376 cells with FL118 does not significantly modulate the expression and phosphorylation of Erk1/2. At the same time, FL118 reduces the expression of pAKT (ser473). (**B**) Treatment of T24 cells with FL118 enhances the phosphorylation of Akt and Erk1/2. (**C**) Treatment of UMUC-3 cells with FL118 downregulated the expression of active AKT and ERK1/2. GAPDH presented in (**A**–**C**) is internal controls for total protein-loading.

**Figure 5 cancers-12-03413-f005:**
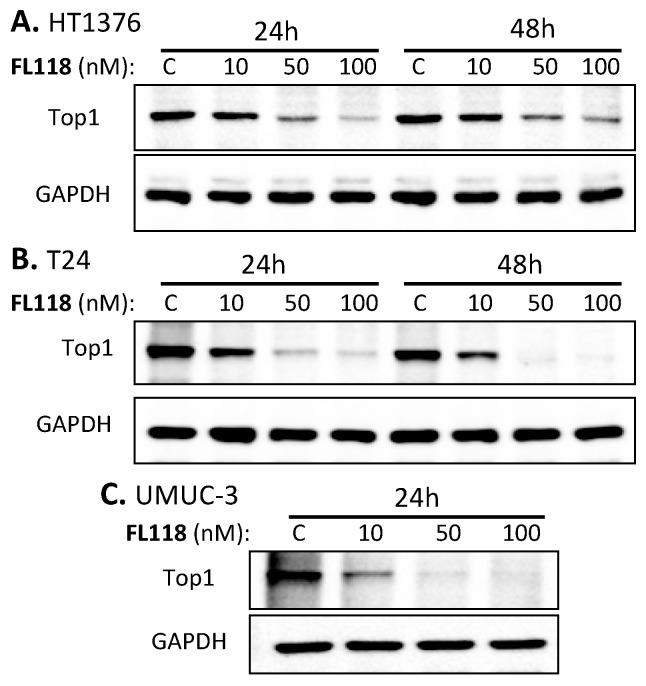
Sensitivity of bladder cancer cells to FL118 is irrelevant to Top1 level after FL118 treatment. Subconfluent bladder cancer cells in 6-well plates were treated with (10, 50, 100 nM) and without (C, vehicle control) FL118 for 24 h and/or 48 h, as shown, followed by Western blot analyses to determine the expression of Top1. (**A**) Treatment of HT1376 cells with FL118 resulted in the least inhibition of Top 1 expression after FL118 treatment. (**B**) Treatment of T24 cells with FL118 resulted in the inhibition of Top 1 expression after FL118 treatment. (**C**) Treatment of UMUC3 cells with FL118 resulted in the most inhibition of Top 1 expression after FL118 treatment. GAPDH presented in (**A**–**C**) is the internal control for total protein-loading.

**Figure 6 cancers-12-03413-f006:**
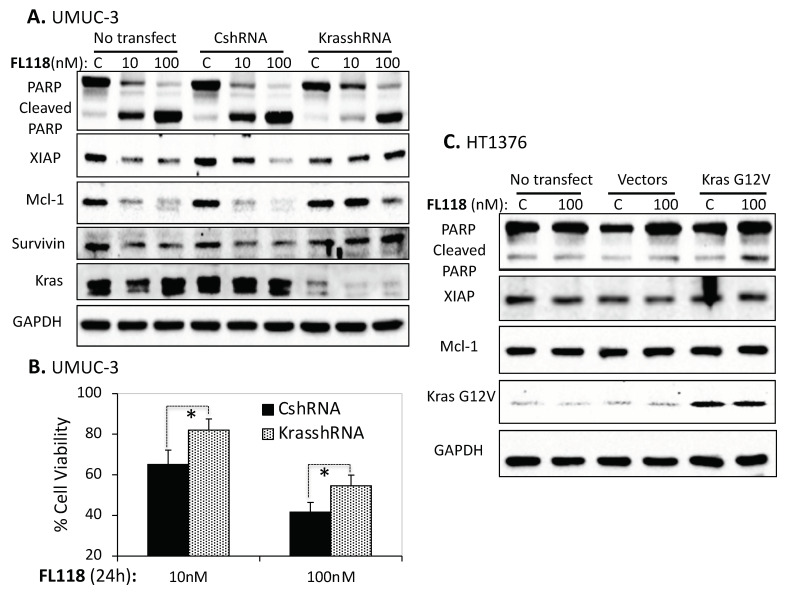
Role of mutant Kras in FL118-mediated apoptosis and inhibition of anti-apoptotic protein expression and cell viability. (**A**) Silencing of Kras G12C via Kras-specific shRNA in UMUC-3 cells increases cell resistance to FL118-induced apoptosis (less PARP cleavage). UMUC-3 cells were infected with Kras-specific lentiviral shRNA identified in our previous studies [21] or control shRNA followed by treatment with DMSO or FL118 (10 and 100 nM) for 24 h. At the end of treatment, Western blot analysis was performed to check PARP cleavage and the expression of anti-apoptotic proteins (XIAP, Mcl-1, survivin). GAPDH was used as an internal control. (**B**) 3-(4,5-dimethylthiazol-2-yl)-2,5-diphenyltetrazolium bromide (MTT) assay was performed to determine cell viability with 10 nM and 100 nM FL118 treatment in UMUC-3 cells infected with control scramble shRNA and Kras-specific shRNA. Each bar in the histogram data is the mean ± SD from three independent assays. (**C**) Effect of forced expression of Kras G12V in HT1376 cells. HT1376 cells were transfected with Kras G12V or control vectors followed by treatment with DMSO or 100 nM FL118 for 48 h. At the end of treatment, Western blot analysis was performed to check the expression of apoptosis-related proteins. GAPDH was used as an internal control of total protein-loading. * *p*-value < 0.05.

**Figure 7 cancers-12-03413-f007:**
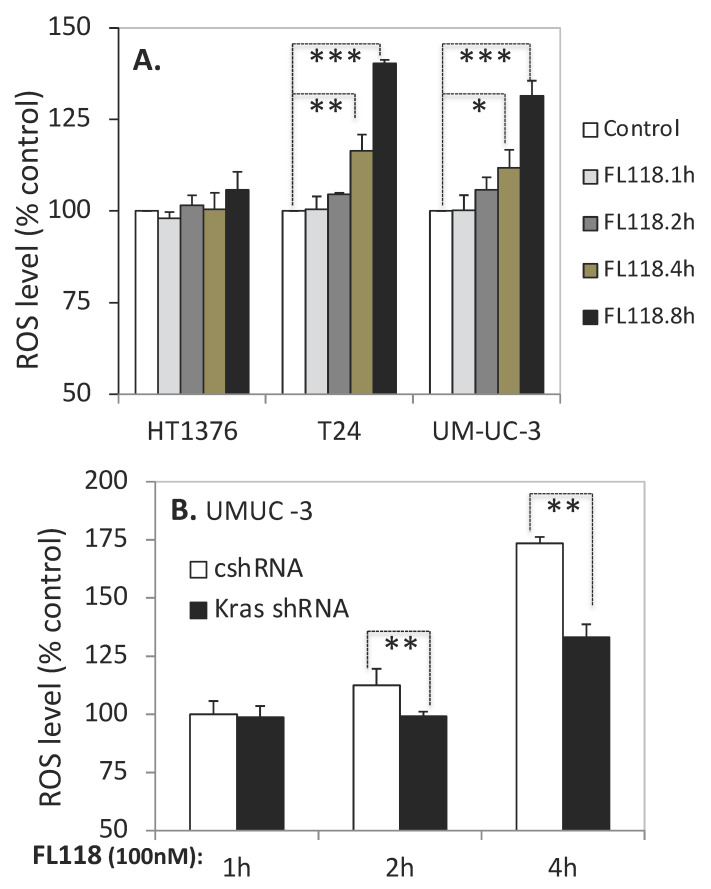
Determination of reactive oxygen species (ROS) production in bladder cancer cells. (**A**) FL118-induced ROS production in bladder cancer cell lines. HT1376, T24 and UMUC-3 cells in 96-well plates were treated with DMSO or 100 nM FL118 for different time points as indicated. At the end of treatment, ROS production was measured using Abcam’s 2′,7′-dichlorofluorescin diacetate (DCFDA) cellular reactive oxygen species detection kit, as described in the Materials and Methods section. Values were shown as mean ± SD derived from three experiments. (**B**) Silencing Kras G12C via Kras-specific shRNA in UMUC-3 cells decreases ROS production after FL118 treatment. UMUC-3 cells were infected with Kras-specific lentiviral shRNA or control shRNA followed by treatment with DMSO or 100 nM FL118 for different time periods. At the end of treatment, ROS production was measured as described in the previous experiment. Values were shown as mean ± SD derived from three experiments. * *p*-value < 0.05, ** *p*-value < 0.01, *** *p*-value < 0.001

**Figure 8 cancers-12-03413-f008:**
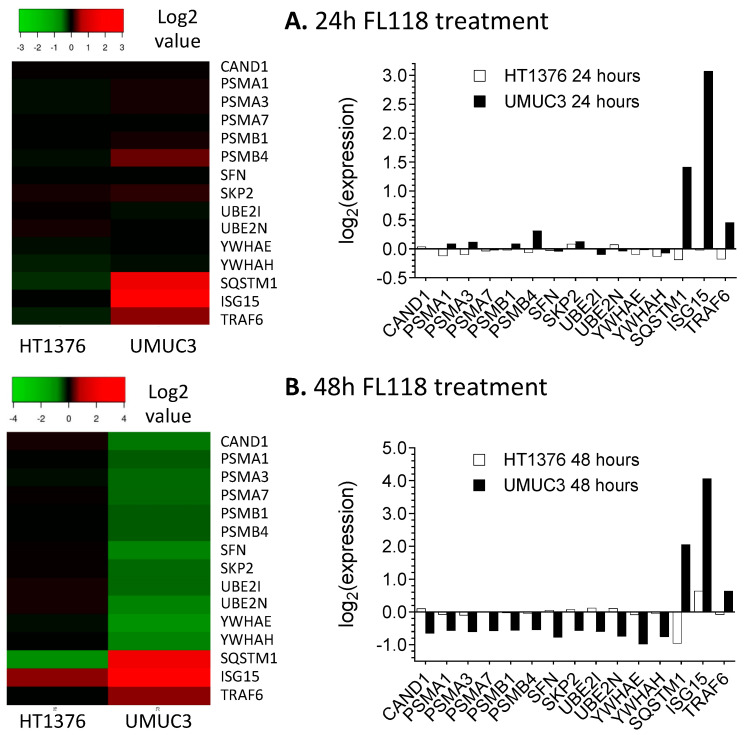
Effects of FL118 on Kras pathway-associated ubiquitination (Ub), de-Ub and proteasome-related proteins. HT1376 and UMUC3 cells were treated with FL118 (20 nM) for 24 h and 48 h. Proteomics analyses were then performed as described in the Method section. The data shown here is the effect of FL118 on the Kras pathway-associated Ub, de-Ub and proteasome-related proteins for 24 h (**A**) and 48 h (**B**). All data in proteomics analyses are in triple replicates in parallel with triple vehicle controls (refer to Appendix A).

**Figure 9 cancers-12-03413-f009:**
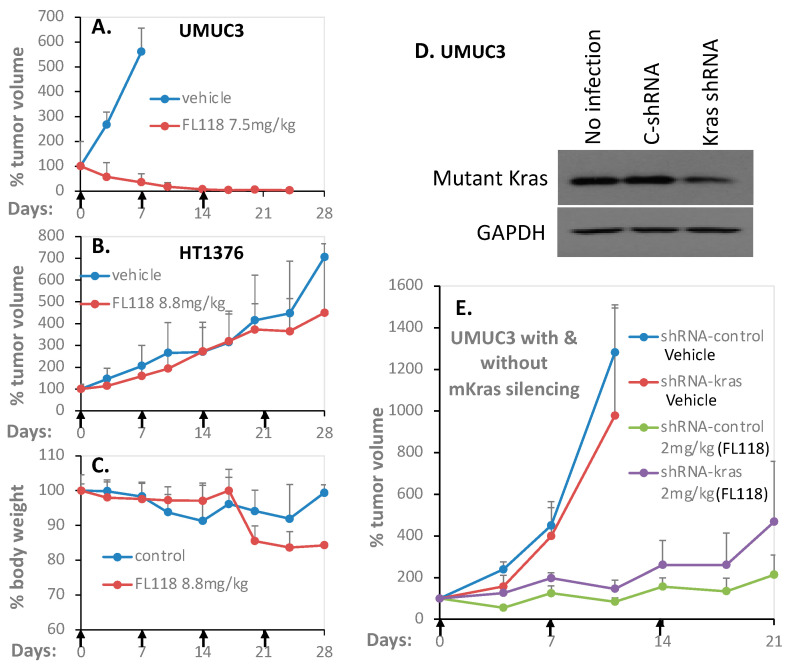
Antitumor efficacy of FL118 in severe combined immunodeficiency (SCID) mice bearing human bladder cancer xenograft tumors. The human bladder cancer xenograft tumors were first generated through implanting bladder cancer cells at the flank area of SCID mice. Then, the tumors were isolated, and individual experimental mice were subcutaneously implanted with 30–50 mg nonnecrotic tumor masses at the flank area of individual mice. Seven to 14 days after tumor transplantation at which the implanted bladder xenograft tumors were grown to 100–150 mm^3^ (defined as day 0), mice were randomly divided into the required groups (5 mice per group) for treatment vial oral administration of FL118 at the doses as shown or vehicles with a schedule of weekly × 3 or 4 (arrowed) as described in the “Materials and Methods” section. (**A**) FL118 efficacy against UMUC-3 bladder cancer cell-established xenograft tumors in SCID mice. (**B**) FL118 efficacy against HT1376 bladder cancer cell-established xenograft tumors in SCID mice. The tumor growth curves shown in A and B are the mean ± SD derived from 5 mice. (**C**) SCID mouse body weight changes after treatment with vehicle or with FL118. The mouse body weight curves shown in C are the mean ± SD derived from 5 mice. (**D**) Western blots validate the expression of mutant Kras in UMUC3 cells in the condition of no infection, with control shRNA lentiviral particle transfection or with Kras shRNA lentiviral particle transfection (Of note, this is the stably infected multiple-infectant mixture, which usually had the issue with some endogenous mutant Kras, due to some unclear reasons). GAPDH is the internal control. (**E**) FL118 exhibited was less effective in inhibiting UMUC3 tumors with a partial silencing of mutant Kras using Kras shRNA. The tumor growth curves shown in E are the mean ± SD derived from 5 mice.

**Table 1 cancers-12-03413-t001:** Relative IC50 comparison among the bladder cancer cell lines used in the studies.

Cell Line	Kras/Hras/Nras Status	IC_50_ *
HT1376	Wild type	>200 nM
T-24	Hras mutation	~13 nM
UMUC3	Kras mutation	~3 nM
RT112	Wild type	>160 nM
253 J B-V	Wild type	>110 nM
TCCSUP (HTB-5)	Wild type	>160 nM
SNU-C2B (CRC)	Kras mutation	~2.7 nM
HCT116 (CRC).	Kras mutation	~0.51 nM

* IC_50_ is the 72 h FL118 treatment.

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
