# Peer review of "Mutant Kras as a Biomarker Plays a Favorable Role in FL118-Induced Apoptosis, Reactive Oxygen Species (ROS) Production and Modulation of Survivin, Mcl-1 and XIAP in Human Bladder Cancer"

_cancers, 2020, doi:10.3390/cancers12113413_

Round 1
Reviewer 1 Report
The authors have positively responded to all my remarks and made the necessary changes to the manuscript.
Reviewer 2 Report
None.
This manuscript is a resubmission of an earlier submission. The following is a list of the peer review reports and author responses from that submission.
Round 1
Reviewer 1 Report
This study reveals the role of FL118 in high-grade bladder cancer, especially Kras mutant bladder cancer cells. The study is of great interest, because of its novelty and very interesting findings with potential clinical impact. Additionally, this manuscript is clearly organized and well described, but it needs minor corrections before accepted for publication.
Comments:
- I suggest to mention tumor xenograft model used for in vitro model.
- Page 14. Section 3.9. I am a little confused about ex vivo experiments. Could you indicate in vitro and ex vivo specific experiments used in that study?
- Page 2. Introduction. What type of cancer cells did you applied in anticancer drug FL118 identifying study (16 ref.)?
- Page 19. Conclusion. It would be interested to point out the FL118 as a potential chemotherapeutic drug for specific types of cancer, according to your previous findings (see above remark). That issue requires better discussion.
- Page 3. Introduction. What is the background of expectation for FL118 to use it in clinical trial in 2021?
- Page 17. Discussion. In my opinion data for FL118 high efficacy in colorectal Kras mutant cells should be published as soon as possible and there is no need to discussed your findings.
- Please, refer to ‘Cancers’ template for the order of sections.
Reviewer 2 Report
- Introduction is lengthy. Should be significantly reduced to most relevant information to current results. other information can be moved to discuss if needed. eg. Ln 86-108
- Ln111-129. seems to summarize the current findings and sound like an abstract within the intro. Should be removed.
- Methods should include all the details of antibodies, vectors etc.... (cat #, dilutions used etc).
- Table 1. IC50 values are based on 48hrs or 72hrs observation?
- Fig 6A. Is it Wild type KRAS that was checked? What about mutant form? include the antibody used in methods.
- In the proteomics data..what was the status of proteins analyzed in earlier figs (eg. Survivin, Xiap, Topo1, Akt, Kras etc etc). Did it correlate with the earlier fig data. Include those figs and discuss.
- What is the direct target of FL118? Include this in intro.